# The Effects of a Brief Lifestyle Intervention on the Health of Overweight Airline Pilots during COVID-19: A 12-Month Follow-Up Study

**DOI:** 10.3390/nu13124288

**Published:** 2021-11-27

**Authors:** Daniel Wilson, Matthew Driller, Paul Winwood, Ben Johnston, Nicholas Gill

**Affiliations:** 1Te Huataki Waiora School of Health, The University of Waikato, Hamilton 3216, New Zealand; 2Faculty of Health, Education and Environment, Toi Ohomai Institute of Technology, Tauranga 3112, New Zealand; paul.winwood@toiohomai.ac.nz (P.W.); nicholas.gill@waikato.ac.nz (N.G.); 3Sport and Exercise Science, School of Allied Health, Human Services and Sport, La Trobe University, Melbourne 3086, Australia; m.driller@latrobe.edu.au; 4Aviation and Occupational Health Unit, Air New Zealand, Auckland 1142, New Zealand; ben.johnston@otago.ac.nz; 5New Zealand Rugby, Wellington 6011, New Zealand

**Keywords:** healthy eating, weight loss, moderate-to-vigorous physical activity, sleep, lifestyle medicine

## Abstract

The aim of this study was to perform a 12-month follow-up of health parameters after a 17-week lifestyle intervention in overweight airline pilots. A parallel-group (intervention and control) study was conducted amongst 72 overweight airline pilots (body mass index > 25) over a 12-month period following the emergence of COVID-19. The intervention group (*n* = 35) received a personalized dietary, sleep, and physical activity program over a 17-week period. The control group (*n* = 37) received no intervention. Measurements for subjective health (physical activity, sleep quality and quantity, fruit and vegetable intake, and self-rated health) via an electronic survey, and objective measures of body mass and blood pressure were taken at baseline and at 12 months. Significant interactions for group × time from baseline to 12-months were found for all outcome measures (*p* < 0.001). Body mass and mean arterial pressure significantly decreased in the intervention group when compared to the control group (*p* < 0.001). Outcome measures for subjective health (physical activity, sleep quality and quantity, fruit and vegetable intake, and self-rated health) significantly increased in the intervention group when compared to the control group (*p* < 0.001). Results provide preliminary evidence that a brief three-component healthy sleep, diet and physical activity intervention can elicit and sustain long-term improvements in body mass and blood pressure management, health behaviors, and perceived subjective health in pilots and may support quality of life during an unprecedented global pandemic.

## 1. Introduction

The COVID-19 pandemic has impacted operations of numerous industries, including the aviation industry which has been significantly disrupted by global travel restrictions, causing a substantial economic decline within the industry [1]. Following the World Health Organization’s characterization of COVID-19 as a pandemic on 11 March 2020, the global commercial airline industry experienced an approximate 60–80% decrease in flight operations during the proceeding months [2]. Accordingly, airline pilots have been affected by decreased work availability [1], job security, financial concerns, increased time spent confined to the indoors due to self-isolation requirements during travel [3], and limited control over food choices during hotel self-isolation after flying internationally. The consequent psychosocial impacts of these conditions may adversely affect the engagement in health promoting behaviors [4].

The COVID-19 pandemic has influenced considerable changes to behavior, and subsequent physical and mental health related outcomes [4]. Authorities in countries worldwide have implemented strict control strategies in attempt to limit the spread of the virus [5]. Consequently, these viral spread mitigation measures in the community pose significant barriers to engagement with health promoting behaviors [6]. For example, financial insecurity, elevated psychosocial stress, and emotional dysregulation may lower motivation and limit accessibility to healthful dietary behaviors [7,8]. Further, stay at home isolation and lockdown measures present an inhibitory effect on engagement in physical activity [4,9].

Negative effects on physical [10] and mental wellbeing, along with elevated levels of psychosocial stress [11] have been reported in research exploring the effects of COVID-19 environmental conditions, such as social distancing and lockdown confinement in adults. Decreases in physical and mental health during COVID-19 have shown associations with unhealthy lifestyle behaviors; sedentary behavior, physical inactivity, poor sleep quality, and unhealthy dietary intake [11]. Prospective cohort studies exploring health behavior status during lockdowns have reported increased sedentary behavior and physical inactivity [12], decreased fruit and vegetable intake [13], increased alcohol intake [8], and increased sleep problems [14], yet little evidence has been reported regarding the prolonged effects after lockdown.

Overweight, obesity and hypertension are independently associated with unhealthy lifestyle behaviors; insufficient sleep, poor diet, and physical inactivity [15,16,17,18]. Widespread societal and economic implications of COVID-19 present perturbations to these health behaviors [3,4,8,10]. Unhealthy lifestyle risk factors synonymous with an elevated risk of non-communicable disease are a risk factor for COVID-19 complications and severity of health outcomes following infection [6]. Markedly, obesity is associated with chronic low-grade inflammation, impaired innate immunity and immunologic compromise [19]. Indeed, recent studies report increased morbidity and mortality risk from COVID-19 in those with obesity [20]. Overweight and obesity are also major risk factors for essential hypertension, of which emerging evidence denotes as a risk factor strongly associated with adverse outcomes from COVID-19 [21].

Behavioral countermeasures for individuals are vital determinants to health resilience amongst exposure to unprecedented environmental events such as the COVID-19 pandemic and its widespread implications [7]. Obtaining seven to nine hours of sleep per night [22], consuming ≥400 g of fruit and vegetables per day fruits and vegetables [23], and engaging in ≥150 min of moderate-to-vigorous physical activity intensity per week are three protective lifestyle behaviors that significantly reduce all-cause mortality [23,24,25], and have a positive effect on physical and mental health [26,27], support healthy bodyweight and blood pressure management [15], and support immune system function [28]. Given the evidence for physical activity, healthy nutrition and sleep quality in promoting health outcomes, it is of public health importance that effective evidence-based interventions targeting the promotion of these behaviors are established for intervention preventive measures to mitigate the adverse health effects of future lockdowns [7].

Our previous research investigated the use of a personalized three-component healthy eating, physical activity and sleep hygiene intervention for promoting health during a COVID-19 lockdown in New Zealand [29]. The intervention’s effectiveness at four-months has been reported [29], which revealed significant improvements in health behavior and subjective health. The aim of the current study is to report on the longer-term outcomes of the intervention; specifically, to evaluate the effects on weight loss and blood pressure. Further, to evaluate what health behavioral changes are sustained or decayed over a period of 12-months and what influence they have on health parameters. It was hypothesized that the intervention group would have significantly greater improvements in health behaviors and health parameters compared to the control group at 12-months. It was also hypothesized that some decay in health behaviors and parameters would be evident in the intervention group from post intervention (4 months) to 12 months.

## 2. Materials and Methods

### 2.1. Design

A two-arm, parallel, controlled design was utilized to evaluate the effectiveness of a brief three-component lifestyle intervention for enhancing and maintaining health behaviors, body mass, and blood pressure management during the COVID-19 pandemic in New Zealand. The acute (17-week) effects of this lifestyle intervention on subjective measures for physical activity, sleep duration, and fruit and vegetable intake have been previously reported [29]. Therefore, the purpose of the present study was to complete a 12-month follow-up to that study [29].

This study was approved by the Human Research Ethics Committee of the University of Waikato in New Zealand; reference number 2020#07. The trial protocol is registered at The Australian New Zealand Clinical Trials Registry (ACTRN12621001105831).

### 2.2. Intervention Timing

After baseline testing, the first five weeks of the intervention period preceded the New Zealand (NZ) Government’s implementation of a four-tier response system to COVID-19 on 21 March 2020 [30]. Thereafter, five weeks were at highest alert level 4, two and a half weeks were at alert level 3, and two weeks were at alert level 2. Thereafter, NZ returned to alert level 1 [31]. Restrictions associated with each alert level is defined elsewhere [29]. Pre-testing occurred between 14 February and 9 March 2020 and follow-up testing was carried out during February and March 2021.

### 2.3. Participants

The study population for both groups consisted of commercial pilots from a large international airline. Inclusion criteria were (a) pilots with a valid commercial flying license, (b) working on a full-time basis, (c) having a body mass index (BMI) of ≥25 (overweight), and (d) a resting blood pressure of >120/80 (systolic/diastolic).

Control group participants consisted of airline pilot volunteers recruited at the time of completing their routine aviation medical examinations located at the airline medical unit during the time of the pre-test period. The intervention group volunteered to participate in the lifestyle intervention by responding to an invitation delivered to all pilots within the company via internal organization communication channels. Participants consisted of pilot rosters including long haul (international flights), short haul (regional flights), and mixed-fleet (variable schedule of regional and short international flights).

All participants provided informed consent prior to participation in the study and were made aware that they could withdraw from the study at any time should they wish to do so. Participants were provided with a unique identification code on their informed consent form, which they were instructed to input into their electronic health survey instead of their name at each data collection timepoint, in order to support anonymity and dataset blinding during data analysis.

The sample size was based on previous research with congruent outcome measures [29]. Clinically significant weight loss is defined as at least a 5% reduction in body mass from the baseline level [32]. Our power calculation suggested that 37 participants were required in each group to achieve an 80% power and 5% significance criterion to detect a 4 kg body mass reduction difference between the intervention and the control. To account for 20% attrition [33], we recruited 89 participants.

### 2.4. Intervention Group

The intervention group participated in a 17-week health intervention consisting of individualized goal setting for physical activity, healthy eating, and sleep hygiene. The intervention commenced with a one-hour individual face-to-face consultation session with an experienced health coach at the airline medical unit. For the intervention group, all participants conducted consultations with the same health coach. In this initial consultation session, the pilots’ barriers and facilitators to health behavior change were assessed with methods outlined elsewhere [34], which were factored into the development of an individualized health program. Further, personalized collaborative goal setting was carried out for the pilot with assistance from the health coach, establishing appropriate outcome, performance, and process goals [35] for (a) sleep hygiene, (b) healthy eating, and (c) physical activity. A mid-intervention phone call was utilized to support adherence, monitor progress and measure compliance to health behaviors. The intervention utilized 20 participant contacts; including 2 face-to-face consultations (baseline and follow-up), 1 telephone call and 17 intra-intervention emails. For full detail of the procedures associated with the intervention readers are referred to the study of Wilson and colleagues [29].

### 2.5. Control Group

The participants in the control group received no intervention or instruction regarding health behaviors during the study timeframe. Pilots were invited to voluntarily complete an electronic survey and consent to providing records of their cardiovascular disease risk factor data from their aviation medical examinations. Pilots who volunteered to participate during the previously defined baseline testing period were sent an invitation via email to voluntarily complete the electronic survey again during the post intervention period and then finally again at the completion of their proceeding annual aviation medical examination to provide insight into the effects of COVID-19 on their health. The control group were invited to participate in the intervention after follow-up testing.

### 2.6. Outcome Measures

Measurements for subjective health (physical activity, sleep quality and quantity, fruit and vegetable intake, and self-rated health) via an electronic survey, and objective measures of body mass and blood pressure were taken at baseline and 12-month follow-up (see Figure 1).

Prior to attending data collection sessions, participants were instructed to avoid any strenuous exercise, stimulants (for example, caffeine or energy drinks), or large meals 4 h before testing. Height was recorded with a SECA 206 height measures and body mass was measured with SECA 813 electronic scales (SECA, Hamburg, Germany). For body mass measurement, participants were wearing clothes with emptied pockets and footwear removed. Blood pressure was measured with an OMRON HEM-757 device (Omron Corporation, Kyoto, Japan), which has been successfully validated independently against international criteria [36]. Measurements of blood pressure were conducted according to the standardized aviation medicine protocol [37]. Systolic blood pressure (SBP) and diastolic blood pressure (DBP) readings were used to calculate mean arterial pressure (MAP) with the following formula: DP + 1/3(SP−DP) [38]. Resting pulse was measured using a Rossmax pulse oximeter SB220 (Rossmax Taipei, Taiwan, China) after a 5-min period of sitting in a chair quietly. All measurement instruments were calibrated prior to data collection.

Outcome measures for subjective health (physical activity, sleep quality and quantity, fruit and vegetable intake, and self-rated health) have been previously described in detail [29]. In brief, moderate-to-vigorous physical activity was determined using the International Physical Activity Questionnaire Short Form (IPAQ) [39]. To measure subjective sleep quality and quantity, the Pittsburgh Sleep Quality Index (PSQI) [40] was utilized. Daily fruit and vegetable intake were measured using dietary recall questions derived from the New Zealand Health Survey [39], and self-rated health was determined using the Short Health Form 12v2 (SF-12v2) [41].

### 2.7. Statistical Analysis

Raw data was extracted from the Qualtrics online survey software (Qualtrics, Provo, UT, USA), entered into an Excel spreadsheet (Microsoft, Seattle, WA, USA) and then imported into the Statistical Package for the Social Sciences (SPSS, version 27; IBM, New York, NY, USA) for all statistical analyses. All variables were assessed using the Shapiro–Wilk’s test (*p* > 0.05) and its histograms, Q-Q plots and box plots for inspection for data normality. Levene’s test was used to test homogeneity of variance. Listwise deletion was applied for individual datasets with missing values or participants who did not complete post-testing.

*t*-Tests were utilized to explore baseline differences between groups. A Chi-squared test was utilized to calculate whether any significant differences exist between fleet types at baseline. A one-way analysis of variance (ANOVA) was utilized to calculate whether any significant differences exist between fleet type for flight frequency and flight hours. Repeated-measures ANOVA using the General Linear Modelling function in SPSS was utilized test for group x time interactions, group effects, and time effects (baseline to 12-months). Age, sex, and flights were included as covariates in the ANOVA. As an additional analysis utilizing paired *t*-test, we examined change in health parameters within the intervention group from post intervention at 4-months to 12-months follow-up. Effect sizes were calculated using Cohen’s *d* to quantify between group effects from pre-testing to post-testing. Effect sizes thresholds were set at >1.2, >0.6, >0.2, <0.2 were classified as *large*, *moderate*, *small*, and *trivial* [42]. The alpha level was set at *p* < 0.05.

## 3. Results

### 3.1. Baseline Characteristics of the Study Population

A total of 143 airline pilots were initially assessed for eligibility and 89 were recruited to participate (see Figure 1). Moreover, 72/89 (81%) pilots (mean ± SD, age; 46 ± 11 year, 11 females, 61 males) provided outcome measure data at all data collection timepoints, which consisted of a combination of short haul, long haul, and mixed fleet rosters (*n* = 28, 35, and 9, respectively). The dropout rates from baseline to 12-months were 17% (ceased employment *n* = 4; testing not fully completed *n* = 3) and 21% (testing not fully completed *n* = 7; ceased employment *n* = 3) for the intervention and control group, respectively.

As displayed in Table 1, at baseline the control and intervention group were of similar height, body mass, DBP, resting pulse, and flight hours. The control group were of advanced age (t(70) = 2.342, *p* = 0.02, *d* = 0.55), consumed more fruit and vegetables (t(70) = 4.570, *p* = <0.001, *d* = 1.08), performed more walking (t(70) = 5.650, *p* = <0.001, *d* = 1.33), higher PCS-12 and MCS-12 scores (t(70) = 7.751, *p* = <0.001, *d* = 1.82, and t(70) = 4.798, *p* = <0.001, *d* = 1.13, respectively), achieved greater sleep duration (t(70) = 3.012, *p* = 0.004, *d* = 0.71), and had a lower MAP (t(70) =−2.598, *p* = 0.011, *d* = 0.61). No significant differences were observed between groups for flights during lockdown and flight hours after lockdown.

### 3.2. Intervention Adherence

For the intervention group, compliance was measured mid-intervention for health behaviors, including average sleep hours, weekly MVPA and daily fruit and vegetable consumption. Thirty-two (91%) were achieving ≥7 h sleep per night and three (9%) were obtaining ≤6.9 h per night. For fruit and vegetable servings per day, 33 (94%) were achieving ≥5 serves of fruit and vegetables per day, whereas two (6%) were eating two to four serves per day. Thirty were achieving ≥150 min MVPA (86%), and five (14%) were completing ≤149 min MVPA per week.

### 3.3. Body Mass, BMI, BP, and Pulse

Group changes from baseline to 12-months are presented in Table 2. Significant interactions for group x time were found for all variables (*p* = <0.001), associated with *small* to *large* effect size differences between groups from baseline to 12-months (see Table 2). The within-group analysis revealed that the intervention elicited significant improvements (*p* < 0.001) in all physical metrics at 12-months, associated with *large* effect sizes (see Table 2). The control group reported a significantly higher body mass and BMI (*p* < 0.001) at 12-months, yet no significant changes were observed in other physical metrics.

### 3.4. Health Behaviors and Self-Rated Health

Significant interactions for group × time were found for all subjective health measures (*p* = <0.001). The within-group analysis reported significantly greater improved health changes from baseline to 12-months for all subjective health measures in the intervention group (*p* < 0.001), associated with *moderate* to *large* effect sizes (see Table 2; Figure 2). In contrast, the control group experienced significant decreases in all outcome measures: sleep duration (t(36) = −2.589, *p* = 0.014, *d* = −0.42), PSQI global score (t(36) = 3.853, *p* = <0.001, *d* = 0.63), and MCS-12 scores (t(36) = −2.300, *p* = 0.027, *d* = −0.38). No significant group differences were reported in other health metrics.

### 3.5. Additional Analysis: Four-Month Post-Intervention to 12-Month Follow-Up Change

Table 3 presents changes within the intervention group between four-months (post-intervention) and 12-months follow-up. There were significant within group differences reported for body mass, BMI, MAP, weekly MVPA (*p* = < 0.05), and DBP (*p* = < 0.001), which were associated with *small* to *moderate* effect sizes towards positive health change. Conversely, a decay of *small* magnitude was observed for health parameters average sleep hours (*d* = −0.23), PCS-12 score (*d* = −0.22), and MCS-12 score (*d* = −0.20). No significant differences were observed for other health parameters.

## 4. Discussion

This is the first 12-month follow-up study after a lifestyle health intervention during the COVID-19 pandemic. The present intervention aimed to improve health-related behaviors and promote healthy changes in bodyweight and blood pressure within overweight pilots through a personalized intervention on healthy eating, sleep hygiene and physical activity. The controlled trial showed that at 12-months follow-up, there appeared to be a significant improvement on health parameters from being provided the 17-week intervention [29], relative to our control group which supports our initial hypothesis. These results are important for researchers and health care professionals to provide insight into prolonged health and quality of life perturbations resulting from COVID-19 that may have potential implications to flight safety. Furthermore, given the dearth of published data pertaining to health behavior interventions during a pandemic and the limited availability of preventive lifestyle-based interventions in pilots, these findings provide novel contributions to this field.

Poor long-term maintenance of weight loss and health behavior change achieved from lifestyle diet and exercise interventions is frequently reported [43]. In our intervention group we observed sustained positive change in health behaviors at 12-months follow-up, relative to baseline characteristics. Further, body mass, blood pressure, and weekly MVPA continued to improve at 12-months compared to post-intervention, whereas other health parameter improvements demonstrated non-significant *trivial* to *small* magnitudes of decay from post intervention. These findings support our secondary hypothesis and are consistent with other health behavior research reporting reduced magnitude of change in health parameters at longitudinal follow-up, compared to post-intervention [44]. A contributing factor that has been proposed is the discontinuation of health care professional support, following intervention completion [45]. Thus, highlighting the importance of ongoing care to facilitate additional health outcome improvements after a brief intervention.

Prospective cohort studies have reported significant increases in body mass within four-months after the onset of the initial COVID-19 lockdown [46,47], yet limited studies have evaluated whether body mass gain is sustained longitudinally after lockdown conditions are lifted. In the present study, participants in the intervention group lost 4.9 kg (↓5.4%), while the control group gained 1.2 kg (↑1.3%) at 12-months, resulting in a 6.1 kg difference in body mass change between groups. Existing literature of lifestyle interventions targeting combined diet, physical activity, and sleep with longitudinal follow-up measures are scarce, limiting comparison accuracy of the present findings to existing research. Airline pilot populations are often male dominant [48]; indeed, our participant sample reflected this demographic. Contrarily, a recent meta-analysis reported the majority of participants in diet and exercise weight management interventions were women [49]. Thus, our study provides important evidence regarding the effectiveness of lifestyle interventions within males.

The intervention utilized 20 participant contacts; including two face-to-face consultations, one telephone call, and 17 intra-intervention emails. Comparatively, a recent review indicated a mean body mass reduction of 2.5 kg at one year follow-up within dietary interventions consisting of 13–24 intra-intervention participant contacts [50]. Another review reported a higher mean body mass reduction of 6.7 kg at one year follow-up pertaining to intensive combined diet and exercise interventions [33]. However, the average length of treatment of these interventions were 37 weeks [33], which is considerably higher than our 17-week intervention [29].

Another gap in the literature base is whether body mass gain observed during lockdown conditions is associated with increased blood pressure, which remains largely unexplored. The body mass gain evident in our control group was associated with a 3.8 mmHg increase in SBP at 12-months, compared to a reduction of 10.3 mmHg observed in the intervention group. The SBP reduction observed in the intervention group is comparable with previous research, which reported a 9.5 mmHg reduction in SBP at 12-months following an intensive diet and exercise lifestyle intervention [51]. Correspondingly, in our intervention group we observed a DBP reduction of 2.9 mmHg, and a further 4.9 mmHg at four-months and 12-months, respectively. Compared with our present findings, a similar longitudinal relationship between body mass and blood pressure following intentional weight loss has been reported [52]. Stevens and colleagues reported participants who succeeded at weight loss maintenance at 36 months post-intervention also maintained blood pressure reduction obtained after the intervention, whereas participants who gained weight also experienced increased blood pressure [52].

We discovered a significantly reduced MCS-12 score and increased PSQI global score (denoting worse sleep) at 12-months follow-up in our control group, with no significant change in other subjective measures. Indeed, previous studies have demonstrated associations between sleep and mental health [53]. Further, negative changes in sleep quality during the pandemic have been associated with negative affect, worry and elevated psychosocial stress [54,55]. These findings may be contributed to by additional factors acting on the pilot population during the time of this study, including decreased work availability [1], job security, financial concerns, increased time spent confined to the indoors due to self-isolation requirements during travel [3], and limited control over food choices during hotel self-isolation after flying internationally.

The magnitude of change observed in the present intervention may be at least partly attributable to; (a) the three-component diet, exercise, and sleep approach, (b) behavioral approaches including collaborative goal setting, face-to-face coaching, telephone call and regular emails, and (c) the potential active interest of the pilot population in enhancing their health to support their aviation medical license. Weight loss factors such as restrictive diets and restrictive caloric patterns have been suggested as effective in the short term, but often have a poor long term success rate, leading to weight regain [56]. Whereas the methods utilized in the present study supported a physically active lifestyle, managing life stress with health behaviors, accountability, and facilitation of autonomy via self-determined goal setting, all of which are associated with successful weight loss maintenance [57]. Airline pilots have been reported to exhibit higher personality scores for maturity, emotional stability, and intelligence when compared to general population norms [58]. These characteristics may positively influence intervention engagement and adherence, thus presenting an important consideration when generalizing our findings to the general population.

Potential limitations of the current study need to be considered in the interpretation of our findings. Firstly, although the sample size provided adequate power to distinguish statistically significant effects in the key outcome variables, the differential recruitment strategies and participant self-selection may have contributed to the differences which were observed at baseline for age, fruit and vegetable intake, weekly walking minutes, PCS-12 and MCS-12 scores, sleep duration, and MAP, with healthier characteristics in favor of the control group. Further, those who voluntarily participated in the intervention may have had strong motivation to engage in healthy change, which may have supported the magnitude of intervention effects observed. Thus, it is advisable that future research implements a randomization design, assigning conditions to participants. Secondly, for feasibility purposes the present study utilized self-report measures for health behaviors, which inherently produce lower accuracy to more invasive objective measures. To enhance outcome measure validity and reliability, utilization of objective methods would be preferential such as actigraphy to monitor sleep and physical activity, and photo logging of dietary behaviors to quantify health behavior metrics; however, this would be somewhat difficult to achieve over a period of 12-months.

## 5. Conclusions

In conclusion, the individualized 17-week healthy eating, physical activity, and sleep hygiene intervention implemented in this study elicited sustained positive change in all outcome measures at 12-months follow-up, relative to baseline characteristics. Further, body mass, blood pressure, and weekly MVPA continued to improve at 12-months compared to post-intervention, whereas other health parameter improvements demonstrated non-significant *trivial* to *small* magnitudes of decay from post intervention. These findings suggest that achievement of these three guidelines promote physical and mental health and improves quality of life among pilots during a global pandemic, yet more regular monitoring post intervention may further strengthen behavior change maintenance. Our study provides preliminary evidence that a multi-behavior intervention may be efficacious during a pandemic and that similar outcomes may be transferrable to other populations. 

## Figures and Tables

**Figure 1 nutrients-13-04288-f001:**
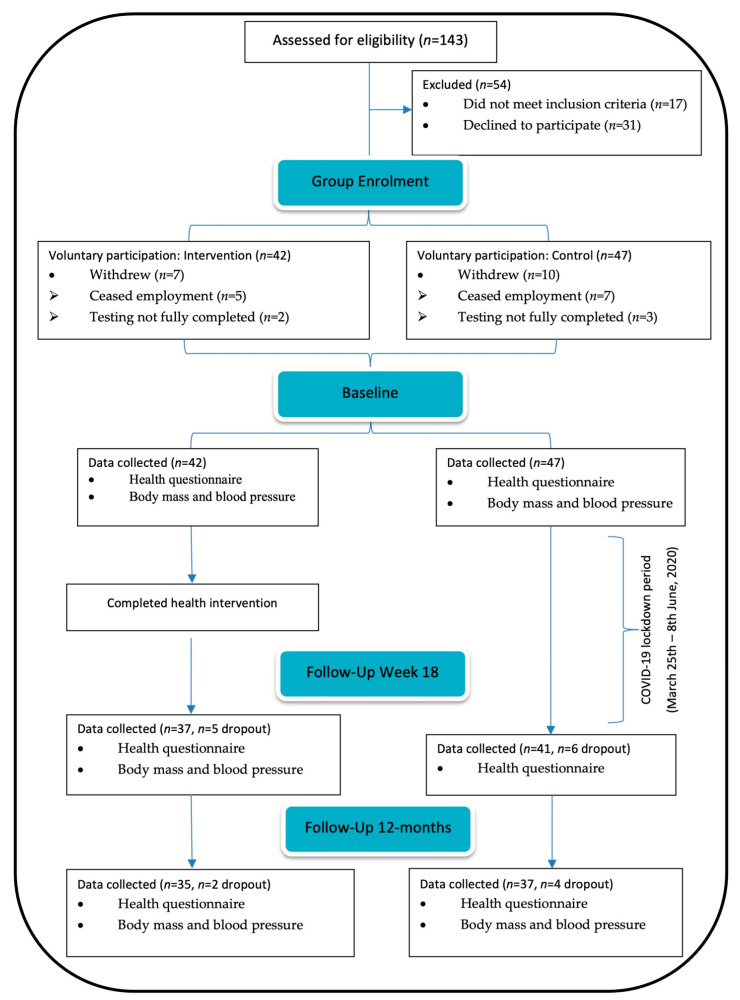
Flow diagram of participant recruitment and data collection.

**Figure 2 nutrients-13-04288-f002:**
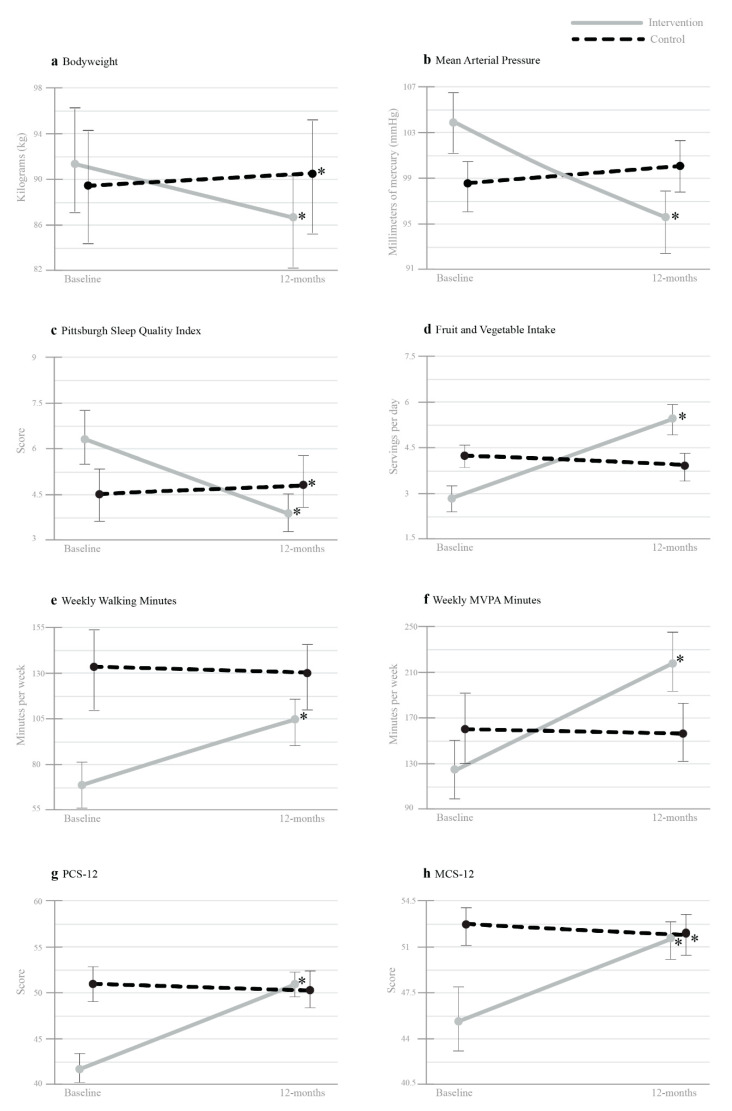
Mean values on objective and subjective health outcomes measured across time (Baseline and 12-months), showing 95% confidence intervals. (**a**), Bodyweight; (**b**), Mean Arterial Presurre; (**c**), Pittsburgh Sleep Quality Indx; (**d**), Fruit and Vegetable Intake, (**e**) Weekly Waliking Minuters; (**f**), Weekly MVPA Minutes; (**g**) PCS-12; (**h**), MCS-12. MVPA = moderate-to-vigorous physical activity. PCS-12 = physical component summary score. MCS-12 = mental component summary score. * indicates a significant difference compared to Baseline.

**Table 1 nutrients-13-04288-t001:** Baseline characteristics of participants.

Parameters	All Participants (*n* = 72)	Intervention (*n* = 35)	Control (*n* = 37)
Sex (f/m)	11/61	7/28	4/33
Age (year)	45.8 ± 11.1	42.8 ± 10.4	48.7 ± 11.2 *
Height (cm)	178.6 ± 7.2	178.5 ± 8.1	178.6 ± 6.3
Body mass (kg)	90.4 ± 13.9	91.7 ± 13.5	89.2 ± 14.5
BMI (kg∙m^2^)	28.3 ± 3.3	28.7 ± 3.3	27.9 ± 2.8
Systolic BP (mmHg)	134.4 ± 11.8	138.4 ± 10.6	130.6 ± 11.7
Diastolic BP (mmHg)	84.8 ± 8.3	86.7 ± 8.1	83.1 ± 8.2
MAP (mmHg)	101.3 ± 8.5	103.9 ± 8.0	98.9 ± 8.5 *
Pulse (bpm)	68.7 ± 9.5	69.2 ± 7.8	68.1 ± 10.9
Hours slept (h/day)	7.3 ± 0.9	7.0 ± 0.8	7.6 ± 0.8 *
IPAQ-walk (min)	102.4 ± 58.5	69.0 ± 37.9	134.0 ± 57.2 *
IPAQ-MVPA (min)	144.5 ± 89.0	125.9 ± 79.7	162.1 ± 94.7
F&V Intake (serve/day)	3.5 ± 1.4	2.8 ± 1.3	4.1 ± 1.1 *
PCS-12 (score)	46.7 ± 6.6	42.1 ± 4.1	51.1 ± 5.5 *
MCS-12 (score)	49.1 ± 7.5	45.3 ± 8.2	52.7 ± 4.5 *
Short Haul (*n*, %)	28 (39%)	20 (57%)	8 (22%) *
Long Haul (*n*, %)	35 (49%)	13 (37%)	22 (59%)
Mixed Fleet (*n*, %)	9 (12%)	2 (6%)	7 (19%)
Flights during lockdown (*n*)	8.0 ± 7.4	7.9 ± 7.7	8.1 ± 7.2
Flight hours after lockdown (h)	152.1 ± 71.9	153.9 ± 63.8	150.5 ± 79.7

Mean ± SD reported for all participants, intervention and control. Abbreviations: SD = standard deviation. BMI—body mass index. BP = blood pressure. MAP = mean arterial pressure. IPAQ = International Physical Activity Questionnaire. MVPA = moderate-to-vigorous physical activity. F&V = fruit and vegetable intake. PCS-12 = physical component summary score. MCS-12 = mental component summary score. * indicates statistical significance between groups (*p* < 0.05). Flight hours after lockdown = flight hours during the 6-months prior to 12-months follow-up testing.

**Table 2 nutrients-13-04288-t002:** Changes in objective and subjective health metrics from baseline and follow-up at 12-months.

	Time (Months)	Intervention			Control			ANOVA (Time × Group Interaction)	Between Group ES
		(*n* = 35)			(*n* = 37)				
		M	SD	Follow-Up Change (95% CI)	M	SD	Follow-Up Change (95% CI)	*p*	*d*
Body mass (kg)	0	91.7	13.5		89.2	14.5			0.2, *Trivial*
	12	86.8	11.3	−4.9 (−3.5–−6.3)	90.5	14.5	1.3 (0.6–1.9)	<0.001	−0.3, *Small*
BMI (kg/m^2^)	0	28.7	3.3		27.9	2.8			0.2, *Trivial*
	12	27.1	2.7	−1.6 (−1.1–−2.0)	28.3	3.7	0.4 (0.2–0.6)	<0.001	−0.4, *Small*
Systolic BP (mmHg)	0	138.4	10.6		130.6	11.7			0.7 *, *Moderate*
	12	128.1	10.3	−10.3 (−7.2–−13.5)	134.4	9.9	3.8 (0.0–7.6)	<0.001	−0.6 *, *Moderate*
Diastolic BP (mmHg)	0	86.7	8.1		83.1	8.2			0.4, *Small*
	12	78.9	8.0	−7.8 (−5.0–−10.6)	83.2	7.0	0.1 (−2.6–2.8)	<0.001	−0.6 *, *Moderate*
MAP (mmHg)	0	103.9	8.0		98.9	8.5			0.6 *, *Moderate*
	12	95.3	7.6	−8.6 (−6.0–−11.3)	100.2	7.2	1.3 (−1.2–4.0)	<0.001	−0.7 *, *Moderate*
Pulse (bpm)	0	69.2	7.8		68.1	10.9			0.1, *Trivial*
	12	63.4	8.5	−5.8 (−3.8–−8.2)	69.2	12.2	1.1 (−0.7–5.6)	<0.001	−0.7 *, *Moderate*
Hours slept (h/day)	0	7.0	0.8		7.6	0.8			−0.7 *, *Moderate*
	12	7.7	0.7	0.7 (0.4–1.1)	7.5	0.7	−0.1 (0.0–−0.3)	<0.001	0.4, *Small*
PSQI Global (score)	0	6.4	2.8		4.5	2.6			0.7 *, *Moderate*
	12	4.1	1.5	−2.3 (−1.7–−3.2)	5.0	2.7	0.5 (0.2–0.7)	<0.001	−0.5 *, *Small*
IPAQ-walk (min)	0	69.0	37.9		134.0	57.2			−1.3 **, *Large*
	12	102.3	69.2	33.3 (22.1–49.6)	122.6	77.6	−11.4 (−16.2–9.4)	<0.001	−0.6 *, *Moderate*
IPAQ-MVPA (min)	0	125.9	79.7		162.1	94.7			−0.4, *Small*
	12	227.0	83.0	101.1 (62.2–126.0)	159.0	99.8	−3.1 (−16.9–11.1)	<0.001	0.8 *, *Moderate*
F&V Intake (serve/day)	0	2.8	1.3		4.1	1.1			−1.1 **, *Moderate*
	12	5.5	1.7	2.7 (2.0–3.2)	3.9	1.3	−0.2 (−0.5–0.1)	<0.001	1.1 **, *Moderate*
PCS-12 (score)	0	42.1	4.1		51.1	5.5			−1.8 **, *Large*
	12	51.7	4.0	9.6 (7.1–10.8)	50.7	4.9	−0.4 (−1.3–0.2)	<0.001	0.1, *Trivial*
MCS-12 (score)	0	45.3	8.2		52.7	4.5			−1.1 **, *Moderate*
	12	51.1	4.9	5.8 (3.6–8.0)	51.8	4.7	−0.9 (−0.1–−1.7)	<0.001	0.2, *Trivial*

Mean ± SD reported for all participants, intervention, and control. Abbreviations: SD = standard deviation. BMI = body mass index. BP = blood pressure. MAP = mean arterial pressure. PSQI = Pittsburgh Sleep Quality Index. IPAQ = International Physical Activity Questionnaire. F&V = fruit and vegetable intake. PCS-12 = physical component summary score. MCS-12 = mental component summary score. * indicates statistical significance between groups (*p* < 0.05). ** indicates statistical significance between groups (*p* < 0.001).

**Table 3 nutrients-13-04288-t003:** Additional analysis: Changes in objective and subjective health metrics from post intervention at 4-months to follow-up at 12-months in the intervention group.

	Time (Months)	Intervention (*n* = 35)			Effect Size
		M	SD	Post-Follow-Up Change (95% CI)	*d*
Body mass (kg)	4	87.7	12.8	-	-
	12	86.8	11.3	−0.97 (−1.81–0.1)	−0.47, *small **
BMI (kg/m^2^)	4	27.5	3.1	-	-
	12	27.1	2.7	−0.32 (−0.58–0.07)	−0.44, *small **
Systolic BP (mmHg)	4	130.9	11.1	-	-
	12	128.1	10.3	−2.89 (−6.09–0.32)	−0.31, *small*
Diastolic BP (mmHg)	4	83.8	9.7	-	-
	12	78.9	8.0	−4.86 (−7.56–−2.15)	−0.62, *moderate ***
MAP (mmHg)	4	99.5	9.4	-	-
	12	95.3	7.6	−4.11 (−6.77–−1.46)	−0.53, *small **
Pulse (bpm)	4	62.6	7.2	-	-
	12	63.4	8.5	0.74 (−2.0–3.5)	0.09, *trivial*
Hours slept (h/day)	4	7.8	1.0	-	-
	12	7.7	0.7	−0.11 (−0.27–0.05)	−0.23, *small*
PSQI Global (score)	4	4.1	1.8	-	-
	12	4.1	1.5	−0.09 (−0.31–0.14)	−0.13, *trivial*
IPAQ-walk (min)	4	94.3	96.5	-	-
	12	102.3	69.2	8.0 (−10.4–26.4)	0.15, *trivial*
IPAQ-MVPA (min)	4	207.6	79.0	-	-
	12	227.0	82.6	19.7 (5.57–33.74)	0.48, *small **
F&V Intake (serve/day)	4	5.6	1.9	-	-
	12	5.5	1.7	−0.13 (−0.49–0.23)	0.12, *trivial*
PCS-12 (score)	4	52.3	4.5	-	-
	12	51.7	4.0	−0.54 (−1.38–0.29)	−0.22, *small*
MCS-12 (score)	4	54.5	5.7	-	-
	12	51.1	4.9	−0.52 (−1.41–0.36)	−0.20, *small*

Mean ± SD reported for the intervention group. Abbreviations: SD = standard deviation. BMI = body mass index. BP = blood pressure. MAP = mean arterial pressure. PSQI = Pittsburgh Sleep Quality Index. IPAQ = International Physical Activity Questionnaire. F&V = fruit and vegetable intake. PCS-12 = physical component summary score. MCS-12 = mental component summary score. * indicates statistical significance (*p* < 0.05). ** indicates statistical significance (*p* < 0.001).

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
