# Peer review of "The Effects of a Brief Lifestyle Intervention on the Health of Overweight Airline Pilots during COVID-19: A 12-Month Follow-Up Study"

_nutrients, 2021, doi:10.3390/nu13124288_

Round 1

Reviewer 1 Report

This is a very interesting brief lifestyle intervention that was undertaken during COVID-19 pandemic, among overweight airline pilots. The current manuscript assessed the effect of the intervention over a period of 12-month follow up and provided evidence on the sustained positive effect of the intervention in health behaviors. The manuscript is well written but requires some improvements and details, specially in the methods section.

Please, see bellow my comments:

  1. I understand that this is a secondary analysis of the lifestyle intervention. However, it is still important to start your ‘study design’ section with a sentence given the design of the intervention (Two-arm intervention….). Also the sentence on line 106 is not clear, please revise it.
  2. There are also information on the lifestyle intervention, measurements and intervention procedures all under the subtitle “design”. I recommend to revise it, and rearrange the information under more coherent subtitles.
  3. Sentences on Lines 118 to 121 should go to introduction.
  4. Please clarify the participants invitation. You mentioned in line 128 that both control and intervention participants are pilots from the same company. In line 134 you said that all pilots from the company received an invitation to participate in the lifestyle intervention. My interpretation of this is that control participants received the invitation to participate in the intervention, but opted not to at the first contact but were later approached and included at the time of completing their routine aviation medical examinations. However, in line 164 you said they were blind to the intervention. Please clarify this.
  5. You only mentioned that the primary objective of this lifestyle intervention is to promote weight loss among pilots in line 144. This should be made clearer earlier in the manuscript.
  6. Table 1 should go to results section.
  7. Please confirm the SD of BMI among controls, as it seems there were included normal weight individuals considering the current SDs (28+-4).
  8. Please clarify if the health coach was the same for all intervention participants.
  9. Line 188 – I suggest you to give more details in this manuscript of the subjective measurement.
  10. In Figure 1 you need to add the lost to follow-up at 18-weeks and 12-month follow-up.
  11. In line 234 you presented the results for compliance, but it was not explained how it was measured in the methods.
  12. In line 317 you gave details of the intervention procedure that were not provided in the methods. Please, move this to methods.
  13. One final comment is related to the percentage of men in both groups. As expected, there were more men than women since the target population were airline pilots. However, most lifestyle interventions tend to recruit more women than men. So this study provide important evidence on the effect of lifestyle intervention in a sample predominantly of men. This should also be discussed in your manuscript.

Author Response

Author Response: Thank you to reviewer 1 for taking the time to review our manuscript. The edits and suggestions provided below have improved the overall quality of the paper. We have made amendments based on your suggestions, and our individual responses to each of your comments are listed below.

Point 1: I understand that this is a secondary analysis of the lifestyle intervention. However, it is still important to start your ‘study design’ section with a sentence given the design of the intervention (Two-arm intervention….). Also the sentence on line 106 is not clear, please revise it.

Response 1: Thank you for pointing out the lack of descriptive detail here, please see the leading sentence of section “2.1 Design” for amended text: “A two-arm, parallel, controlled design was utilized to evaluate the effectiveness of a brief three-component lifestyle intervention for enhancing and maintaining health behaviours, body mass and blood pressure management during the COVID-19 pandemic in New Zealand.”

Line 106 sentence has been removed and replaced by amended text.

Point 2: There are also information on the lifestyle intervention, measurements and intervention procedures all under the subtitle “design”. I recommend to revise it, and rearrange the information under more coherent subtitles.

Response 2: Thanks for this suggestion to improve writing coherency. The information indicated has been rearranged within the methods section to address this.

Point 3: Sentences on Lines 118 to 121 should go to introduction

Response 3: Thanks for picking this up, this sentence was already in the introduction and was duplicated on line 118-121. This has been removed from methods and it remains within the introduction as you have recommended.

Point 4: Please clarify the participants invitation. You mentioned in line 128 that both control and intervention participants are pilots from the same company. In line 134 you said that all pilots from the company received an invitation to participate in the lifestyle intervention. My interpretation of this is that control participants received the invitation to participate in the intervention, but opted not to at the first contact but were later approached and included at the time of completing their routine aviation medical examinations. However, in line 164 you said they were blind to the intervention. Please clarify this.

Response 4: Thanks for noting this inconsistency in explanation, your interpretation was correct. The statement that the control group were blind to the intervention has been removed to correct this.

Point 5: You only mentioned that the primary objective of this lifestyle intervention is to promote weight loss among pilots in line 144. This should be made clearer earlier in the manuscript.

Response 5: Additional information has been added in the introduction on lines 94-95. Amended sentence in full: “The aim of the current study is to report on the longer-term outcomes of the intervention; specifically, to evaluate the effects on weight loss maintenance. Further, to evaluate what health behavioral changes are sustained or decayed over a period of 12-months and what influence they have on health parameters.”

Point 6: Table 1 should go to results section.

Response 6: Table 1 has been moved to the results section as recommended.

Point 7: Please confirm the SD of BMI among controls, as it seems there were included normal weight individuals considering the current SDs (28+-4).

Response 7: Thanks for noticing this typo, the SD was indeed 3 (28±3) – this has been corrected in Table 1.

Point 8: Please clarify if the health coach was the same for all intervention participants.

Response 8: Yes, the health coach was the same for all intervention participants. A sentence has been added to provide this detail on lines 170-171.

Point 9: Line 188 – I suggest you to give more details in this manuscript of the subjective measurement.

Response 9: Thanks for the suggestion, we have now added further information on lines 227-232 to expand detail on subjective measurements. Added information: “Outcome measures for subjective health (physical activity, sleep quality and quantity, fruit and vegetable intake, and self-rated health) have been previously described in detail [29]. In brief, moderate-to-vigorous physical activity was determined using the International Physical Activity Questionnaire Short Form (IPAQ) [39]. To measure subjective sleep quality and quantity, the Pittsburgh Sleep Quality Index (PSQI) [40] was utilized. Daily fruit and vegetable intake were measured using dietary recall questions derived from the New Zealand Health Survey [41], and self-rated health was determined using the Short Health Form 12v2 (SF-12v2) [42].”

Point 10: In Figure 1 you need to add the lost to follow-up at 18-weeks and 12-month follow-up.

Response 10: Thanks, this detail has been added to Figure 1.

Point 11: In line 234 you presented the results for compliance, but it was not explained how it was measured in the methods.

Response 11: To support clarification of this issue, further detail has been provided on lines 186-187. “A mid-intervention phone call was utilized to support adherence, monitor progress and measure compliance to health behaviours”.

Point 12: In line 317 you gave details of the intervention procedure that were not provided in the methods. Please, move this to methods.

Response 12: Thanks for this insight, this detail has been integrated into the methods section on lines 178-180.

Point 13: One final comment is related to the percentage of men in both groups. As expected, there were more men than women since the target population were airline pilots. However, most lifestyle interventions tend to recruit more women than men. So this study provide important evidence on the effect of lifestyle intervention in a sample predominantly of men. This should also be discussed in your manuscript.

Response 13: Thanks for this suggestion, we have added further information on lines 366-370. Added paragraph: “Airline pilot populations are often male dominant [49]; indeed, our participant sample reflected this demographic. Contrarily, a recent meta-analysis reported the majority of participants in diet and exercise weight management interventions were women [50]. Thus, our study provides important evidence regarding the effectiveness of lifestyle interventions within males.”.

Reviewer 2 Report

This study reported physical changes and changes in subjective symptoms after one year of a 17-week intervention for sleep, diet, and physical activity. Although the short-term effects of the intervention were already reported by the authors, the present study suggests that the effects persisted after the intervention ended, making it a valuable study. The manuscript was straightforward throughout and very well prepared.

Still, there are several minor corrections that should be made, as shown below.

Terms such as "Short or Long Haul" and "Mixed fleet" do not make much sense to readers who are not familiar with the aviation industry, so please provide a simple explanation.

In line 188 to 190, the authors stated that “Outcome measures for subjective health (physical activity, sleep quality and quantity, fruit and vegetable intake, and self-rated health) have been previously described.” This is unfriendly to the readers. It would be nice to refer to the prior literature for more details, but a simple explanation of the subjective measures used in this study: SF-12, PSQI, IPAQ, sleep habits, and fruit and vegetable intake would be appreciated.

Related to the above, there is no description of abbreviations such as MAP, IPAQ, and MVPA.

The intervention group in this study were those who participated voluntarily. They also had low levels of baseline mental and physical health. Therefore, it should be emphasized that there is a possibility that pilots with a strong motivation to become healthy participated in the study, which may have led to such robust results.

Generally, it's not easy to become a pilot. They must be the chosen ones who have gone through many trials and tribulations. Therefore, it should be mentioned that it may be difficult to directly apply the results of this study to the general population. For example, there is a report on the differences in personality traits between airplane pilots and the general population (see https://pubmed.ncbi.nlm.nih.gov/12841441/).

Author Response

Author Response: Thank you to reviewer 2 for taking the time to review our manuscript. The edits and suggestions provided below have improved the overall quality of the paper. We have made amendments based on your suggestions, and our individual responses to each of your comments are listed below.

Point 1: Terms such as "Short or Long Haul" and "Mixed fleet" do not make much sense to readers who are not familiar with the aviation industry, so please provide a simple explanation.

Response 1: We have added additional detail on lines 130-132. “Participants consisted of pilot rosters including long haul (international flights), short haul (regional flights), and mixed-fleet (variable schedule of regional and short international flights).”

Point 2: In line 188 to 190, the authors stated that “Outcome measures for subjective health (physical activity, sleep quality and quantity, fruit and vegetable intake, and self-rated health) have been previously described.” This is unfriendly to the readers. It would be nice to refer to the prior literature for more details, but a simple explanation of the subjective measures used in this study: SF-12, PSQI, IPAQ, sleep habits, and fruit and vegetable intake would be appreciated.

Related to the above, there is no description of abbreviations such as MAP, IPAQ, and MVPA.

Response 2: Thanks for pointing out this lack of clarity, we have now added further information on lines 227-232 to expand detail on subjective measurements. Abbreviation was provided for MAP on line 218. Amended paragraph: “Outcome measures for subjective health (physical activity, sleep quality and quantity, fruit and vegetable intake, and self-rated health) have been previously described in detail [29]. In brief, moderate-to-vigorous physical activity was determined using the International Physical Activity Questionnaire Short Form (IPAQ) [39]. To measure subjective sleep quality and quantity, the Pittsburgh Sleep Quality Index (PSQI) [40] was utilized. Daily fruit and vegetable intake were measured using dietary recall questions derived from the New Zealand Health Survey [41], and self-rated health was determined using the Short Health Form 12v2 (SF-12v2) [42].”

Point 3: The intervention group in this study were those who participated voluntarily. They also had low levels of baseline mental and physical health. Therefore, it should be emphasized that there is a possibility that pilots with a strong motivation to become healthy participated in the study, which may have led to such robust results.

Response 3: Thank you for this insight, we have added further information to lines 424-426 to emphasize this consideration. Added paragraph: “Further, those who voluntarily participated in the intervention may have had strong motivation to engage in healthy change, which may have supported the magnitude of intervention effects observed.”

Point 4: Generally, it's not easy to become a pilot. They must be the chosen ones who have gone through many trials and tribulations. Therefore, it should be mentioned that it may be difficult to directly apply the results of this study to the general population. For example, there is a report on the differences in personality traits between airplane pilots and the general population (see https://pubmed.ncbi.nlm.nih.gov/12841441/).

Response 4: Thanks for your insight regarding this factor. We have now added additional information on lines 413-416, and we have integrated the reference you provided. Added paragraph: “Airline pilots have been reported to exhibit higher personality scores for maturity, emotional stability, and intelligence when compared to general population norms [57]. These characteristics may positively influence intervention engagement and adherence, thus presenting an important consideration when generalizing our findings to the general population.”

-------------------------
